# Microbiome and Pregnancy Dysbiosis: A Narrative Review on Offspring Health

**DOI:** 10.3390/nu17061033

**Published:** 2025-03-15

**Authors:** Valentina Biagioli, Mariarosaria Matera, Luca Antonio Ramenghi, Raffaele Falsaperla, Pasquale Striano

**Affiliations:** 1Department of Neurosciences, Rehabilitation, Ophthalmology, Genetics, Maternal and Child Health, University of Genoa, 16126 Genoa, Italy; lucaramenghi@gaslini.org (L.A.R.); pstriano@unige.it (P.S.); 2Usl Toscana Sud Est, Pediatric Clinical Microbiomics Service, Misericordia Hospital, Via Senese 161, 58100 Grosseto, Italy; mariarosaria.matera@uslsudest.toscana.it; 3Neonatal Intensive Care Unit, Department Mother and Child, IRCCS Istituto Giannina Gaslini, 16147 Genoa, Italy; 4Department of Medical Science-Pediatrics, University of Ferrara, 44124 Ferrara, Italy; raffaele.falsaperla@unife.it; 5IRCCS Istituto Giannina Gaslini, 16147 Genoa, Italy

**Keywords:** fetal programming, maternal immune activation, maternal microbiome, microbiota–gut–brain axis, pregnancy dysbiosis

## Abstract

**Background:** Emerging evidence suggests that the maternal microbiome plays a crucial role in shaping fetal neurodevelopment, immune programming, and metabolic health. Dysbiosis during pregnancy—whether gastrointestinal, oral, or vaginal—can significantly influence pregnancy outcomes and long-term child health. **Materials and Methods**: The search was performed using databases such as PubMed, Scopus, and Google Scholar including research published from January 2000 to January 2025. The keywords used were “Fetal Programming”, “ Maternal Immune Activation”, “Maternal microbiome”, “Microbiota–Gut–Brain Axis”, and “Pregnancy Dysbiosis”. **Results**: The maternal microbiome undergoes substantial changes during pregnancy, with alterations in microbial diversity and function linked to conditions such as gestational diabetes, obesity, and preeclampsia. Pregnancy-related dysbiosis has been associated with adverse neurodevelopmental outcomes, including an increased risk of autism spectrum disorder (ASD), attention-deficit/hyperactivity disorder (ADHD), and cognitive impairments in offspring. **Conclusions:** Understanding the intricate relationship between maternal microbiota and fetal health is essential for developing targeted interventions. Personalized microbiome-based strategies, including dietary modifications and probiotic supplementation, hold promise in optimizing pregnancy outcomes and promoting health in offspring.

## 1. Introduction

The gut microbiota represents a complex ecosystem of bacteria, viruses, fungi, and protozoa, capable of establishing relationships not only between one microorganism and another (cross-feeding mechanisms), but also with the host thanks to symbiotic and commensalism [1]. This large functional unit, defined as the holobiont, not only defines our state of health but also represents the inheritance that can be transmitted in the mother–fetus dyad [2]. Maternal intestinal and vaginal microbiota are currently the focus of several scientific studies intending to better understand how variations in the microbial consortium can determine the state of health and disease in the offspring. Furthermore, there is a strong interest in understanding the role of the maternal microbial consortium in mediating prenatal factors and determining the fertility and implantation of the embryo. This relevance also applies to the paternal side; scientific studies highlight how the seminal microbiota can influence sperm quality and quantity, and how alterations could cause prostatitis and male infertility [3].

Moreover, during pregnancy, physiological hormonal changes aimed at fetal development and growth, such as reduced peripheral sensitivity to insulin, increased gluconeogenesis, and lipolysis, lead to an imbalance in glycemic balance, with an increase in blood glucose levels [4]. At the same time, butyrate-producing bacteria are reduced during the third trimester of pregnancy, and proteobacteria and pro-inflammatory cytokines are increased [5]. This condition leads to increased intestinal permeability, greater diffusion, and loss of compartmentalization of the bacteria in the intestinal epithelium [6,7]. Therefore, pregnancy dysbiosis, not only intestinal but also oral and in the other microbial niches, is little investigated or not promptly assessed. Moreover, oral dysbiosis is a condition of periodontal disease and correlates with adverse obstetric outcomes such as premature birth [8]. Contreras A. and Herrera JA et al. showed how women with preeclampsia had significantly higher rates of periodontal disease and chronic periodontitis (*p* < 0.001). Furthermore, the bacteria *P. gingivalis* and *E. corrodens* were more represented in the group of women with preeclampsia compared to the control group with full-term delivery [9]. Therefore, a study conducted by Vuong et al. on pregnant mouse models has shown how the dysbiotic picture induced by antibiotic therapy leads to a reduced expression of genes related to axogenesis in the offspring [10]. For these reasons, taking care of a pregnant woman who presents a dysbiotic picture is fundamental to ensure an optimal course of pregnancy and the health of the future unborn child.

Particularly in Westernized societies, where people live in urbanized areas, give excessive attention to hygiene practices, and consume a Western-type diet, consisting of excess simple sugars, complex carbohydrates, and reduced intake of fiber and indigestible carbohydrates, as well as where agrochemicals, intensive farming, and livestock farming are excessively used, there has been a progressive distancing from interactions with the environment and the rhizosphere, thus causing a consequent impoverishment of microbial biodiversity [11,12]. Numerous studies indicate that there is a higher prevalence of allergic and immune-related diseases among children today. Additionally, during their reproductive years, including during pregnancy, women are exhibiting increasingly dysbiotic conditions that are frequently associated with health issues such as obesity and gestational diabetes [13,14,15,16].

This review explores the complex interactions in the mother–fetus–microbiota triad and how they can influence health in fetal growth.

## 2. Materials and Methods

The search was performed using PubMed, Scopus, and Google Scholar databases, including studies published from January 2000 to January 2025. The keywords used were “Fetal Programming”, “ Maternal Immune Activation”, “Maternal microbiome”, “Microbiota–Gut–Brain Axis”, and “Pregnancy Dysbiosis Boolean operators (AND, OR) were used to refine the search results. The articles included were in English and include meta-analyses, systematic literature reviews, and original articles; articles such as case reports, letters, and commentaries were excluded. Additionally, studies on human cohorts and animals were included. The initial studies included 200 (150 from PubMed, 40 from Google Scholar, and 10 from Scopus). For each keyword, according to our search criteria for “Fetal programming”, we found 20 articles; for the keyword “Maternal Immune Activation”, we found 45 articles; for the keyword “Maternal microbiome”, we found 36 articles; for the keyword “Microbiota–gut–brain axis”, we found 70 articles; and for the keyword “Pregnancy Dysbiosis”, we found 29 articles. Two reviewers independently reviewed abstracts; after checking the texts’ eligibility, discrepancies between reviewers were discussed through engagement and discussion with a third reviewer. Through the implemented search strategy, 200 articles were initially identified. After excluding duplicates and studies with similar content, the final selection of articles included in our review was determined.

## 3. How Gut Microbiota Changes During Pregnancy

### 3.1. Gastrointestinal Microbiota

Scientific studies have shown how fecal microbiota transplantation, typical of the first three months of pregnancy, in germ-free mice induces a significant increase in weight and an increase in pro-inflammatory cytokines [17]. In women with overweight and obesity during pregnancy, there is an increase in pro-inflammatory macrophages and in LPS at the level of placental tissue [18]; furthermore, the dysbiosis that characterizes obesity involves an altered production of SCFAs with a reduction in SCFA-producing bacteria (*Faecalibacterium prausnitzii*), thus compromising the regulation of energy metabolism by worsening insulin resistance and hyperglycemia by triggering fetal metabolic syndrome and impacting fetal brain development [19]. Furthermore, women living in low/middle-income countries, with a high rate of poverty and food insecurity and a higher probability of contracting enteric diseases due to poor environmental hygiene, are more likely to manifest environmental enteric dysfunction (EED), a subclinical syndrome in which there is a compromised intestinal barrier, a high degree of inflammation, and, consequently, malabsorption problems [20]. It has been shown that children born to mothers with EED manifest compromised neurological and immune development, both the result of maternal malnutrition. A study on murine models has shown how a state of malnutrition in future mothers shows a significant correlation with EED in the offspring and an impoverishment of SIgA in both mothers and offspring [21]. A study conducted by Gough EK et al. (2021) has shown that changes in the intestinal microbiota of pregnant women in Zimbabwe are a more accurate predictor of birth weight than gestational age [22].

### 3.2. Endometrial Microbiota

For decades, it has been claimed that the placenta and endometrium are sterile organs. Today, thanks to next-generation sequencing (NGS) techniques, several studies are in agreement that the dominant species are Lactobacillus (>70%), followed by *Bifidobacterium*, *Corynebacterium*, *Streptococcus*, and *Staphylococcus* [23]. Furthermore, the endometrium is a site where several players in the immune system work in balance to promote blastocyst implantation during pregnancy [24]. An imbalance in the microbial and immune response in the endometrial niche has been associated with poor obstetric outcomes, including a risk of preterm birth and miscarriage [25]. Moreover, a study by Cicinelli et al. demonstrated that in women with chronic endometritis, there is a reduction in the biodiversity of the endometrial microbial consortium and an increase in pro-inflammatory factors (high TNF-α and IFN-γ levels) [26]. In addition, Yant et al. demonstrated a significant reduction in the rate of preterm birth in pregnant CD1 mice, previously injected intrauterinely with LPS and subsequently administered *Lactobacillus rhamnosus* peritoneally [27].

### 3.3. Vaginal Microbiota

The vaginal microbiota accompanies and evolves throughout all phases of a woman’s life, from puberty to pregnancy to menopause [28]. The microbiota in eubiosis in healthy women of Caucasian ethnicity is represented by a prevalence of *lactobacilli*, which can metabolize glycogen into lactic acid, maintaining the vaginal pH at 3.8–4.4 [29]. In normal pregnancies, the vaginal microbial consortium tends to remain stable except before delivery, when an increase in biodiversity, similar to that in the non-pregnant state, is observed, suggesting that it is a precipitating factor in the onset of labor [30]. However, in cases of vaginal dysbiosis with a microbial imbalance and an increase in pro-inflammatory cytokines, it is associated with adverse obstetric outcomes and a risk of preterm birth [31]. Therefore, several studies show that an increase in microbial biodiversity, with low levels of *Lactobacillus* and an increase in *Mycoplasma* and *Garnerella vaginalis*, occurs in women who have experienced a preterm birth; on the contrary, the presence of *Lactobacillus crispatus* appears to be a protective factor [32].

## 4. Cross-Talk Between Immune System and Microbiota During Pregnancy

### 4.1. Maternal Immune Activation and Intestinal Dysbiosis

Changes in the microbial, hormonal, and metabolic cycle during pregnancy can predispose individuals to a non-physiological dysbiotic picture and dysregulation of the immune system; this condition is known as maternal immune activation (MIA).

MIA is a process of activation of the immune system of pregnant women that involves an alteration of cytokine signaling, epigenetic modifications, and changes in placental function (transport of nutrients and oxygen) and has recently been considered a triggering factor in fetal neurodevelopment and the appearance of behavioral disorders in offspring (autism spectrum disorder (ASD), schizophrenia, attention-deficit/hyperactivity disorder (ADHD), and cognitive/behavioral impairments) [33,34,35,36]. Moreover, epidemiological and experimental studies in animal models have shown that it is precisely the dysregulated activation of the maternal immune response, and not a specific pathogen, that represents a risk factor for neurodevelopmental disorders [37].

In addition to an infectious trigger, MIA can also be induced by acute and chronic inflammatory conditions [38,39,40], such as chronic inflammatory bowel disease (IBD), autoimmune diseases, asthma [41], and low-grade systemic inflammation, like excess weight and obesity [42], gestational diabetes [43], and periodontal disease [44], as well as exposure to stress and polluting and toxic environmental factors [45]. Recently, a correlation has also seemed to emerge between MIA and negative outcome on assisted reproductive technology (ART) [46].

All these conditions are united by intestinal dysbiosis which, through an imbalance of bacterial metabolites, in particular short-chain fatty acids (SCFAs), plays an essential role with respect to the control of barrier permeability and pro-inflammatory cytokines that can trigger and/or amplify the immune activation of MIA. SCFAs are the main microbial metabolites derived from the fermentation of dietary fiber, and butyrate in particular plays an important role in neuro-immuno-endocrine regulation. Butyrate has local effects in the gut, ranging from maintaining the integrity of the intestinal barrier to the production of mucus to protection against inflammation [47]. Butyrate is also able to cross the blood–brain barrier (BBB) and modulate neurodevelopment by interacting directly with neuronal and glial cells but also by inhibiting histone deacetylase (HDAC) activity, thus promoting the acetylation of lysine residues present in nucleosomal histones of nervous system cells [48].

Growing evidence suggests that exposure to pro-inflammatory environmental insults in the intrauterine time window, when offspring are particularly vulnerable to neurodevelopmental issues, results in epigenetic changes [49]. Epigenetic changes, through histone modifications, DNA methylation, and microRNA expression, alter the expression of genes involved in the regulation of proper neuronal growth and migration, dendritic development [50], and synapse formation and function [51]. The altered expression in the brain of immune molecules (cytokines and major histocompatibility complex (MHCI)) also modifies synaptic plasticity and connectivity circuits between the various brain areas [52].

MIA is characterized by an increase in pathogen-associated molecular models (PAMPs) and damage-associated molecular models (DAMPs) but also microbe-associated molecular models (MAMPs) that activate Toll-like receptors on maternal immune cells and placental cells, leading to a huge production of pro-inflammatory cytokines (IL-6, IL-1β, and TNF-α) precisely in this time window of neuro-fetal vulnerability [53]. Inflammatory cytokines cross the placental barrier and reach the fetus [54]. IL-6 stimulates placental activation of Th17 cells, contributing to placental dysfunction. IL-6, TNF-α, and Th17 activate the immune cells residing at the placental level with a further increase in the inflammatory circulation and transfer of the maternal systemic inflammatory state to the fetus. Finally, inflammatory cytokines in the fetus cross the BBB, causing neuroinflammation and interference with the activation and functional expression of microglial cells [55].

### 4.2. Maternal Dysbiosis and Neurodevelopmental Disorders

Experimental mouse models of MIA show a close relationship between maternal gut dysbiosis, intestinal barrier permeability, inflammatory status, and altered neurodevelopment in offspring [56]. This may occur because the microbiota and the central nervous system are interconnected through the microbiota–gut–brain axis by means of bacterial metabolites such as tryptophan and SCFAs or MAMPs, which, in addition to modulating the peripheral immune system, also act with respect to the activation of microglia in the developing brain [57].

Microglia are cells of the innate immune system that can act as intermediaries of microbiota. They respond to stimulation by PAMPs, DAMPs, and MAMPS, secreting pro-inflammatory cytokines including TNF-α, IL-1β, and IL-6 [58].

Erny D et al. showed that the maternal microbiota is crucial for the maturation and activation of microglia. In fact, they observed that in germ-free mice, microglia show morphological, molecular, and spatial abnormalities with increased density in various brain regions and altered response to proteins involved in the regulation of brain development such as CSF1R, CD31, and the activation marker F4/80 [59].

In addition, the phenotypic and functional changes in microglia are closely linked to SCFAs. In fact, in SPF mice, constitutively lacking the SCFA receptor FFAR2, microglia show an aberrant phenotype like that of germ-free mice or those treated with antibiotics that are known to induce dysbiosis with perturbation of microbial diversity. Vuong, H.E et al. confirm that antibiotic-induced dysbiosis in pregnant mice, as well as pregnancy in germ-free mice, leads to reduced expression of axogenesis-related genes, while microbial consortium colonization is able to prevent such abnormalities [60]. Human studies also confirm that infections and antibiotic use during pregnancy correlate with an increased risk of neurodevelopmental complications in offspring [61].

Moreover, it has also recently been shown that dietary tryptophan is metabolized by the gut microbiota into metabolite agonists of the aryl hydrocarbon receptor (AHR) that act directly on microglia and astrocytes that express AHR by mimicking inflammation of the central nervous system [62].

Maternal malnutrition, by inducing alteration of the microbiota of pregnant mice, is also associated with a reduction in white matter in the brains of offspring mice, and studies conducted in children have also associated maternal malnutrition during pregnancy with gut dysbiosis and dysregulation of the axogenesis process [63]. Recently, a two-hit theory has been developed (Estes and Mcallister), according to which MIA represents the first hit with respect to the increase in susceptibility to neurodevelopmental disorders in children [64]. According to this theory, neurodevelopment risk can be amplified by the intervention of a second environmental hit that could occur in the postnatal period [65]. We could therefore conclude that an intervention aimed at preserving maternal eubiosis could play a decisive role in calming MIA through the modulation of barrier permeability, inflammatory state, and microglial functional expression, thus preserving fetal neurodevelopment (Figure 1) [66,67,68].

### 4.3. Diet During Pregnancy and Practical Applications of Clinical Microbiomics

Nutrition during pregnancy is a fundamental element that not only ensures correct fetal growth and maternal well-being but also drives the enrichment (or otherwise) of the intestinal microbiota of the mother and the newborn [69]. Heather A. Paul et al. demonstrated that in mouse models fed a high-fat/sucrose diet for 10 weeks, females were significantly heavier than in the lean control group, showed higher plasma glucose and insulin levels, and showed significantly lower levels of peptide-YY (PYY) [70]. In addition, the gut microbiota resulted in a lower relative abundance of fecal Bifidobacterium spp. and a higher abundance of *Clostridium* [71]. Furthermore, because of HFD and maternal dysbiosis, molecular changes may occur at the fetal level and in the brain of the offspring, such as a reduction in the expression of brain-derived neurotrophic factor (BDNF); the latter promotes neuronal differentiation and regulates learning, memory, and behavior [70]. On the contrary, a diet rich in fiber and polyunsaturated fatty acids (PUFAs) in obese women during pregnancy is associated with an improvement in the microbial community, better bacterial biodiversity, and a reduction in serum zonulin [72]. Moreover, in the MAMI-MED cohort study, conducted by an Italian research group, the dietary patterns of 667 pregnant women were investigated to evaluate their association with birth weight for gestational age. The group of women with a dietary pattern based on ultra-processed foods, such as French fries, sauces, and sweets and an excess of refined complex carbohydrates, showed a greater probability of large-for-gestational-age (LGA) births compared to women belonging to the group with a healthy diet based on fish, white meat, and abundant quantities of fruits and vegetables (OR = 2.213; 95%CI = 1.047–4.679; *p* = 0.038) [73]. Moreover, the SPRING (PRobiotics IN Gestational Diabetes) randomized controlled trial study showed that a cohort of pregnant women with a vegetarian diet had an abundance of *Holdemania*, *Roseburia*, and *Lachnospiraceae*, which are among the major strains producing SCFAs [74]. Furthermore, a cross-sectional study by M. Selma-Royo et al. showed that a maternal diet rich in saturated (SFAs) and monounsaturated fatty acids (MUFAs) was positively associated with an increase in *Firmicutes* in the neonatal microbiota and inversely related to a diet rich in fiber and plant proteins [75].

Following what has been reported, a paradigm shift is therefore necessary in addressing the issue of maternal diet; to date, in line with the present scientific evidence on the importance of microbiota in pregnancy, it is insufficient to adopt the same dietary patterns for all women. An individualized and personalized approach is necessary with the possibility of investigating intestinal microbiota in the first months of pregnancy to be able to act in terms of real health prevention [76]. It will therefore be useful to reduce simple sugars by favoring complex carbohydrates, which are rich in fiber, vitamins, and minerals [77,78], unsaturated fats and fermented foods such as yogurt, kefir, tempeh, and kimchi, which can improve microbial biodiversity, to enable a butyrate-producing microbial profile [79,80].

### 4.4. New Frontiers in Probiotic and Prebiotic Supplementation

The use of supplements during pregnancy is not to be considered the only therapeutic alternative for women with pregnancy dysbiosis but is a useful aid once an individualized nutritional plan has been implemented. Therefore, many probiotics are useful not only for the treatment of intestinal dysbiosis but also as mood modulators [81]. Tian et al. showed that in mouse models, supplementation of *B. longum* subsp. *infantis* E41 and *B. breve* M2CF22M7 significantly improved 5-hydroxytryptophan (5-HTP) secretion, with a positive impact on the regulation of mood and depressive behavior in pregnancy [82]. Moreover, in rats, the administration of *L. farciminis* appears to have a modulatory effect on the hypothalamic–pituitary–adrenal axis (HPA), with downregulation of circulating ACTH and consequently glucocorticoids such as cortisol [83]. Cortisol, in addition to being known as the “stress hormone”, increases intestinal permeability [84,85]. Meanwhile, *Akkermansia muciniphila* instead forfeits intestinal permeability in favor of the turnover of the mucolayer and contrasts a migration of LPS with a mechanism of metabolic endotoxemia; this Gram-negative bacterium belongs to the phylum *Verrucomicrobia* and has the task of degrading intestinal mucus. This activity overstimulates goblet cells to produce more mucus, favoring their turnover [86,87,88]. Furthermore, studies in the literature have shown how in murine models, *A. muciniphila*, thanks to its wall component Amuc1100, can interact with the Toll-like receptor (TLR) found at the apical level of enterocytes; this bacterium–receptor relationship favors the expression of genes that transcribe for proteins that favor tight junctions (occludin and claudin), thus promoting intestinal barrier activity [89]. Therefore, *A. muciniphila* and its metabolites (butyrate and propionate) have been shown to significantly improve the symptoms of preeclampsia in rat models by promoting trophoblast invasion and M2 polarization of macrophages in the placenta [90]. Moreover, Carole Brosseau et al. demonstrated how in pregnant mice, inulin and galactooligosaccharide (GOS) supplementation during gestation increased the *Bacteroidetes*/*Firmicutes* ratio in favor of *Bacteroidetes*, resulting in increased production of acetate detectable in amniotic fluid, and following prebiotic supplementation, B and T cells increased in the fetus and remained in later stages of life also [91].

Future studies will have to follow an approach that is now commonly known as precision medicine. The key today is not a “one-size-fits-all” approach but rather the individualization of pro/pre/postbiotics starting from an accurate analysis of the clinical history, nutritional regime, and enterotype.

## 5. Conclusions

In conclusion, numerous scientific studies have shown that the gestation period is associated with multiple immunological, metabolic, and hormonal changes. These changes are accompanied by alterations in the intestinal microbial consortium of the pregnant woman. To date, however, the processes underlying these changes are not yet well understood. Therefore, new studies are needed to expand our knowledge on how to best preserve health in the maternal–fetal dyad, starting with a consideration of pre- and perinatal factors. Studying the interactions between the microbiota and pregnant women could open new avenues for the identification of early biomarkers of neonatal health.

## Figures and Tables

**Figure 1 nutrients-17-01033-f001:**
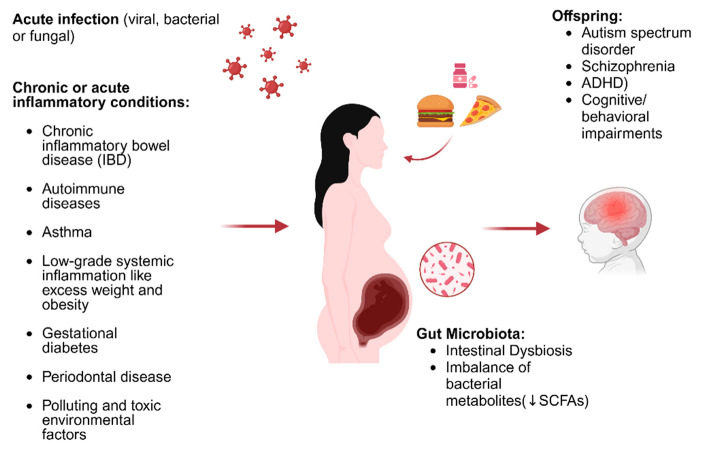
Illustration of the triggers of MIA and how these affect neurodevelopment (edited using biorender.com). ↓: decrease.

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
