# Peer review of "Microbiome and Pregnancy Dysbiosis: A Narrative Review on Offspring Health"

_nutrients, 2025, doi:10.3390/nu17061033_

Round 1

Reviewer 1 Report

Comments and Suggestions for Authors

  1. This paper describes a literature review (not sure which type, and that needs to be very clear) relating the maternal microbiome to neurodevelopmental outcomes in the offspring.  My concerns below really deal with the design of the review (?type) and the organization and discussion of the findings.
  2. There needs to be more details regarding the data base search; please detail the exact databases used and the exact search strategy for each.  These could be put into an appendix. Also in the appendix should be a listing of the included studies, and their characteristics.
  3. I am confused as to why other meta analyses and systematic literature reviews were included here.  They could be used to inform the identification of studies, but to my understanding this is not a "review of reviews" and thus cannot mix original research with such reviews.  Please revise.
  4. I understand that duplicates were properly removed, but was is meant by similar studies?
  5. The results/discussion are confusing as to the point of this review.  While the title of the paper promises a review of the evidence that maternal dysbiosis is associated with neurodevelopmental outcomes in the offspring, there is really little done here (a bit, but little).
  6. Further there is no conceptual model...is the dysbiosis in the mother due to diet (that is alluded to in the discussion), environmental factors, or other exposures that may be preventable.  Or are the authors contending that the dysbiosis is what is amenable to preventive efforts?  So in the context of intervention and prevention, when does it occur?
  7. Further, there is no evidence clearly provided that the maternal microbiome is associated with neurodevelopment in the offspring...only a bit of animal evidence in the offspring.  Indeed, the authors argue that a pro-inflammatory environment, likely due to poor diet (leading to poor microbiome) is the intermediate variable in the pathway.  A diagram like figure 1 (or figure 1 could be modified) showing the pathways would be helpful as a roadmap for the discussion.

Author Response

Comments 1: This paper describes a literature review (not sure which type, and that needs to be very clear) relating the maternal microbiome to neurodevelopmental outcomes in the offspring.  My concerns below really deal with the design of the review (?type) and the organization and discussion of the finding

Response 1: Thanks for your comments. The type of the review is a Narrative review. 

Comments 2: There needs to be more details regarding the data base search; please detail the exact databases used and the exact search strategy for each.  These could be put into an appendix. Also in the appendix should be a listing of the included studies, and their characteristics.

Response 2: We are agree and we added more information about databases and Boolean operators. 

Comments 3: I understand that duplicates were properly removed, but was is meant by similar studies?

Response 3: The titles and abstracts were independently reviewed by two reviewers to identify potentially relevant articles. Discrepancies between the reviewers were resolved through consultation with a third reviewer.

Comments 4: The results/discussion are confusing as to the point of this review.  While the title of the paper promises a review of the evidence that maternal dysbiosis is associated with neurodevelopmental outcomes in the offspring, there is really little done here (a bit, but little).

Response 4: Thanks for your comments. We are agree with that. We are now improve the evidence. Unfortunately the are a few clinical study than animals. 

Comments 5: Further there is no conceptual model...is the dysbiosis in the mother due to diet (that is alluded to in the discussion), environmental factors, or other exposures that may be preventable.  Or are the authors contending that the dysbiosis is what is amenable to preventive efforts?  So in the context of intervention and prevention, when does it occur?

Response 5: We have seen that an alteration of the pregnancy microbiota is completely physiological; however, when there is a significant alteration accompanied by dysmetabolism, GDM, and obesity which are accompanied by a systemic inflammatory state, they hurt the microbial/maternal/fetal health condition, for this reason we believe that acting from a preventive perspective through an evaluation of the microbiota and the dietary regime are necessary to improve fetal well-being and the future unborn child.

Comments 6: Further, there is no evidence clearly provided that the maternal microbiome is associated with neurodevelopment in the offspring...only a bit of animal evidence in the offspring.  Indeed, the authors argue that a pro-inflammatory environment, likely due to poor diet (leading to poor microbiome) is the intermediate variable in the pathway.  A diagram like figure 1 (or figure 1 could be modified) showing the pathways would be helpful as a roadmap for the discussion.

Response 6: Thank for your point. Additional evidence were added  were possible to discuss the correlation. 

Reviewer 2 Report

Comments and Suggestions for Authors

1
Title:
Maternal Microbiome and Neurodevelopment: The Impact of Pregnancy Dysbiosis on Offspring Health
The title should identify if the study reports (human or animal) trial data or is a systematic review, meta-analysis, or replication study.
https://www.mdpi.com/journal/nutrients/instructions#preparation
Abstract
I am unsure whether this is a systematic or narrative review, although the Methods suggest that it is a narrative review.
While the Abstract mentions the databases and keywords used for research, it does not provide any information on how the studies were selected or the criteria for inclusion. More transparency is needed here.
1. Introduction
The Introduction addresses multiple aspects but may improve in terms of structural coherence. The shifts between concepts occasionally appear sudden, hindering readers' grasp of the overarching narrative.
The Introduction mentions the limited investigation into pregnancy dysbiosis, which is important. Elaborating on existing literature and clearly defining specific gaps in knowledge would strengthen the argument for further research.
The Introduction cites various studies and findings but does not provide a thorough explanation or context for several of these assertions.
Please avoid phrases such as “we know “-line 45.
Please remove “with” in line 54.
The following sentence reads odd “Numerous studies show how today in the pediatric age there is a greater presence of allergic and immune-related diseases and how in women in the fertile and pregnant period there is increasingly a dysbiotic picture with often related pathological conditions such as obesity and gestational diabetes [8,9].” Please revise. Perhaps the Authors want to use the following instead: “Numerous studies show indicate that there is a higher prevalence of allergic and immune-related diseases among children today. Additionally, women during their reproductive years, including pregnancy, are exhibiting increasingly dysbiotic conditions that are frequently associated with health issues such as obesity and gestational diabetes [8,9].” Lines 58-62.
Lines 58-62. The Authors write ““Numerous studies” and cite only 2 papers. Please add more references.
“Future studies “should not be mentioned in the Introduction.
2
The Introduction does not provide a clear and explicit statement of the objectives or questions that the review addresses. While it discusses the significance of the maternal microbiota in fetal health and touches upon the implications of dysbiosis, it does not specifically outline the precise aims of the review or the questions it seeks to answer. An effective Introduction typically includes a statement clearly defining what the Authors intend to achieve with the review. This could involve outlining specific hypotheses being tested, questions related to the impact of the maternal microbiome on offspring neurodevelopment, or the importance of addressing dysbiosis during pregnancy.
There are too few cited works in this section.
I think that after “et al.” there should be a year of publication in parentheses. Please revise the entire manuscript.
2. Materials and Methods
Lines 68-69: Please rewrite “The search was performed using databases such as PubMed, Scopus, and Google Scholar, including searches published from January 2000 to January 2025. “ Perhaps this could be used instead: “The search was performed using PubMed, Scopus, and Google Scholar databases, including searches published from January 2000 to January 2025.”
This does not read well: “The studies initially included were 200 (150 from PubMed, 40 from Google Scholar, and 10 from Scopus).” Lines 74-75. Perhaps this could be used instead: “The initial studies included 200 (150 from PubMed, 40 from Google Scholar, and 10 from Scopus).”
This does not read well: “ Through the established search strategy, 200 articles were found, subsequently duplicates and similar studies were excluded leading to include in our review. “Lines 79-81. Please replace with “Through the implemented search strategy, 200 articles were initially identified. After excluding duplicates and studies with similar content, the final selection of articles included in our review was determined.”
Why did the Authors not write about the search terms' combinations or Boolean operators that were employed?
“are “-please replace with “were”. The Methods section is typically written using past tenses.
The inclusion and exclusion criteria are somewhat vague.
There is no mention of how data was extracted from the selected articles or whether a systematic review protocol was followed, which is crucial for ensuring objectivity and minimizing bias. Actually, this current work does not seem to be a systematic review, but still- how was bias going to be minimized?
Why was the timeline from January 2000 to January 2025, specifically?
3
3. Results
3.1. How Gut Microbiota changes during pregnancy
3.1.1. The Gastrointestinal Microbiota
There should be a comma after “lipolysis” in line 91.
The following does not read well “At the same time, during the third trimester of pregnancy, butyrate-producing bacteria are reduced and there is an increase in proteobacteria and pro-inflammatory cytokines [12].” Please replace it with “At the same time, butyrate-producing bacteria are reduced during the third trimester of pregnancy, and proteobacteria and pro-inflammatory cytokines are increased [12]. ” Lines 92-94.
There should be a comma before “and loss” in line 95.
There should be a comma after “transplantation” in line 99.
The following does not read well “A study conducted by Gough EK et al., has shown that in pregnant women in Zimbabwe, changes in the intestinal microbiota are a more accurate predictor of birth weight than gestational age [19]. “ Maybe the Authors want to use the following instead: “A study conducted by Gough EK et al. (2021) has shown that changes in the intestinal microbiota of pregnant women in Zimbabwe are a more accurate predictor of birth weight than gestational age [19].”
3.1.2. The Endometrial microbiota
Please remember about the year of publication after “et al.”
3.1.3. The vaginal microbiota
Please remove the comma before “in healthy” in line 97.
Please remove the comma after “ethnicity” in line 134.
There is an issue with commas in the following: “In normal pregnancies, the vaginal microbial consortium tends to remain stable except before delivery, when an increase in biodiversity like the non-pregnant state suggests that it is a precipitating factor in the onset of labor [27].” Lines 135-138. Please rewrite: “In normal pregnancies, the vaginal microbial consortium tends to remain stable except before delivery, when an increase in biodiversity, like the non-pregnant state, suggests that it is a precipitating factor in the onset of labor [27].“
4. Discussion
4.1. Maternal Immune Activation and Intestinal dysbiosis
Why the references are written [30,31,32,33] are not written like this: [30-33]? Line 155. Please follow the journal’s instructions: In the text, reference numbers should be placed in square brackets [ ], and placed before the punctuation; for example [1], [1–3]
https://www.mdpi.com/journal/nutrients/instructions#references
4
Please replace “with respect to “ with “concerning” in line 153.
There are many issues with commas and incorrect grammatically sentences. Please improve the English language.
Please avoid phrases such as “we speak of “-line 149.
The Authors say that MIA is universally acknowledged due to acute infections. This may risk oversimplifying the complexity of how various immune triggers can lead to MIA.
4.2. Maternal dysbiosis and Neurodevelopmental disorders
“figure” should be capitalized-line 239.
English must be revised, the same the use of commas. This disrupts the flow.
There is an issue with references “[61,62,63]” in line 239. Please see my comment above.
4.2. Diet during pregnancy and practical applications of clinical microbiomics
I noticed that the same numbering, “4.2,” is used for different subheadings (Maternal dysbiosis and Neurodevelopmental disorders, Diet during pregnancy and practical applications of clinical microbiomics, New frontiers in probiotic and prebiotic supplementation).
4.2. New frontiers in probiotic and prebiotic supplementation
The research primarily references animal studies (mouse and rat models). This section does not clarify which data are human data.
Where are future directions?
5. Conclusions
Please do not use question marks. Rewrite, please.
This paragraph is not conclusive at all.
General remarks:
Does the research primarily focus on Westernized societies? This is how I understood this in the Introduction. The manuscript might not be generalizable to other populations if this is true.
The manuscript often explores links between dysbiosis and negative outcomes, but determining transparent causal relationships remains difficult. More longitudinal studies would be required to establish causality conclusively. I do not think that the Authors include longitudinal data.
5
The microbiome is a complex and dynamic system, and attributing specific outcomes to dysbiosis may oversimplify its role. The Authors should be concerned about other factors.
Although the study recognizes the need for further research, it could provide more detail on specific knowledge gaps or the methodologies that would be most effective in driving progress in the field.
The recommendation for dietary modifications and probiotic supplementation assumes a uniform response among all pregnant women. Would this hold true for each individual's variances in microbiota composition?
I am very surprised that this review contains only 82 references. This is a small number for a review.
The manuscript is not engaging.

Comments on the Quality of English Language

English Language must be revised.

Author Response

Comments 1: Title:
Maternal Microbiome and Neurodevelopment: The Impact of Pregnancy Dysbiosis on Offspring Health
The title should identify if the study reports (human or animal) trial data or is a systematic review, meta-analysis, or replication study.
https://www.mdpi.com/journal/nutrients/instructions#preparation
Abstract
I am unsure whether this is a systematic or narrative review, although the Methods suggest that it is a narrative review.
While the Abstract mentions the databases and keywords used for research, it does not provide any information on how the studies were selected or the criteria for inclusion. More transparency is needed here.
1. Introduction
The Introduction addresses multiple aspects but may improve in terms of structural coherence. The shifts between concepts occasionally appear sudden, hindering readers' grasp of the overarching narrative.
The Introduction mentions the limited investigation into pregnancy dysbiosis, which is important. Elaborating on existing literature and clearly defining specific gaps in knowledge would strengthen the argument for further research.
The Introduction cites various studies and findings but does not provide a thorough explanation or context for several of these assertions.
Please avoid phrases such as “we know “-line 45.
Please remove “with” in line 54.
The following sentence reads odd “Numerous studies show how today in the pediatric age there is a greater presence of allergic and immune-related diseases and how in women in the fertile and pregnant period there is increasingly a dysbiotic picture with often related pathological conditions such as obesity and gestational diabetes [8,9].” Please revise. Perhaps the Authors want to use the following instead: “Numerous studies show indicate that there is a higher prevalence of allergic and immune-related diseases among children today. Additionally, women during their reproductive years, including pregnancy, are exhibiting increasingly dysbiotic conditions that are frequently associated with health issues such as obesity and gestational diabetes [8,9].” Lines 58-62.
Lines 58-62. The Authors write ““Numerous studies” and cite only 2 papers. Please add more references.
“Future studies “should not be mentioned in the Introduction.

Response 1: Thanks a lot for your detailed comment. We are proceded to revised entire introduction. 

Comments 2: The Introduction does not provide a clear and explicit statement of the objectives or questions that the review addresses. While it discusses the significance of the maternal microbiota in fetal health and touches upon the implications of dysbiosis, it does not specifically outline the precise aims of the review or the questions it seeks to answer. An effective Introduction typically includes a statement clearly defining what the Authors intend to achieve with the review. This could involve outlining specific hypotheses being tested, questions related to the impact of the maternal microbiome on offspring neurodevelopment, or the importance of addressing dysbiosis during pregnancy.There are too few cited works in this section.
I think that after “et al.” there should be a year of publication in parentheses. Please revise the entire manuscript.2. Materials and Methods Lines 68-69: Please rewrite “The search was performed using databases such as PubMed, Scopus, and Google Scholar, including searches published from January 2000 to January 2025. “ Perhaps this could be used instead: “The search was performed using PubMed, Scopus, and Google Scholar databases, including searches published from January 2000 to January 2025.”
This does not read well: “The studies initially included were 200 (150 from PubMed, 40 from Google Scholar, and 10 from Scopus).” Lines 74-75. Perhaps this could be used instead: “The initial studies included 200 (150 from PubMed, 40 from Google Scholar, and 10 from Scopus).”This does not read well: “ Through the established search strategy, 200 articles were found, subsequently duplicates and similar studies were excluded leading to include in our review. “Lines 79-81. Please replace with “Through the implemented search strategy, 200 articles were initially identified. After excluding duplicates and studies with similar content, the final selection of articles included in our review was determined.”
Why did the Authors not write about the search terms' combinations or Boolean operators that were employed?“are “-please replace with “were”. The Methods section is typically written using past tenses.
The inclusion and exclusion criteria are somewhat vague. There is no mention of how data was extracted from the selected articles or whether a systematic review protocol was followed, which is crucial for ensuring objectivity and minimizing bias. Actually, this current work does not seem to be a systematic review, but still- how was bias going to be minimized?Why was the timeline from January 2000 to January 2025, specifically?

Response 2: Thanks for your point. We are reorganized better this part of the review. The type of the review is a Narrative review. The timeline was selected for choice the more recent literature. 

Comments 3: Results
3.1. How Gut Microbiota changes during pregnancy
3.1.1. The Gastrointestinal Microbiota
There should be a comma after “lipolysis” in line 91.
The following does not read well “At the same time, during the third trimester of pregnancy, butyrate-producing bacteria are reduced and there is an increase in proteobacteria and pro-inflammatory cytokines [12].” Please replace it with “At the same time, butyrate-producing bacteria are reduced during the third trimester of pregnancy, and proteobacteria and pro-inflammatory cytokines are increased [12]. ” Lines 92-94.
There should be a comma before “and loss” in line 95.
There should be a comma after “transplantation” in line 99.
The following does not read well “A study conducted by Gough EK et al., has shown that in pregnant women in Zimbabwe, changes in the intestinal microbiota are a more accurate predictor of birth weight than gestational age [19]. “ Maybe the Authors want to use the following instead: “A study conducted by Gough EK et al. (2021) has shown that changes in the intestinal microbiota of pregnant women in Zimbabwe are a more accurate predictor of birth weight than gestational age [19].”
3.1.2. The Endometrial microbiota
Please remember about the year of publication after “et al.”
3.1.3. The vaginal microbiota
Please remove the comma before “in healthy” in line 97.Please remove the comma after “ethnicity” in line 134.There is an issue with commas in the following: “In normal pregnancies, the vaginal microbial consortium tends to remain stable except before delivery, when an increase in biodiversity like the non-pregnant state suggests that it is a precipitating factor in the onset of labor [27].” Lines 135-138. Please rewrite: “In normal pregnancies, the vaginal microbial consortium tends to remain stable except before delivery, when an increase in biodiversity, like the non-pregnant state, suggests that it is a precipitating factor in the onset of labor [27].

Response 3: Thank for your correction. We are rewrote this part better. 

Comments 4: Discussion
4.1. Maternal Immune Activation and Intestinal dysbiosis
Why the references are written [30,31,32,33] are not written like this: [30-33]? Line 155. Please follow the journal’s instructions: In the text, reference numbers should be placed in square brackets [ ], and placed before the punctuation; for example [1], [1–3]
https://www.mdpi.com/journal/nutrients/instructions#references
4
Please replace “with respect to “ with “concerning” in line 153.
There are many issues with commas and incorrect grammatically sentences. Please improve the English language.
Please avoid phrases such as “we speak of “-line 149.
The Authors say that MIA is universally acknowledged due to acute infections. This may risk oversimplifying the complexity of how various immune triggers can lead to MIA.
4.2. Maternal dysbiosis and Neurodevelopmental disorders
“figure” should be capitalized-line 239.
English must be revised, the same the use of commas. This disrupts the flow.
There is an issue with references “[61,62,63]” in line 239. Please see my comment above.
4.2. Diet during pregnancy and practical applications of clinical microbiomics
I noticed that the same numbering, “4.2,” is used for different subheadings (Maternal dysbiosis and Neurodevelopmental disorders, Diet during pregnancy and practical applications of clinical microbiomics, New frontiers in probiotic and prebiotic supplementation).
4.2. New frontiers in probiotic and prebiotic supplementation
The research primarily references animal studies (mouse and rat models). This section does not clarify which data are human data.
Where are future directions?

Response 4: thank you, we are highlighted better and more insightful the future direction and and we proceeded to make the requested changes. 

Comments 5: Conclusions
Please do not use question marks. Rewrite, please.
This paragraph is not conclusive at all.
General remarks:
Does the research primarily focus on Westernized societies? This is how I understood this in the Introduction. The manuscript might not be generalizable to other populations if this is true.
The manuscript often explores links between dysbiosis and negative outcomes, but determining transparent causal relationships remains difficult. More longitudinal studies would be required to establish causality conclusively. I do not think that the Authors include longitudinal data.
5
The microbiome is a complex and dynamic system, and attributing specific outcomes to dysbiosis may oversimplify its role. The Authors should be concerned about other factors.
Although the study recognizes the need for further research, it could provide more detail on specific knowledge gaps or the methodologies that would be most effective in driving progress in the field.
The recommendation for dietary modifications and probiotic supplementation assumes a uniform response among all pregnant women. Would this hold true for each individual's variances in microbiota composition?
I am very surprised that this review contains only 82 references. This is a small number for a review.
The manuscript is not engaging.

Response 5: We thank you for the great revision work carried out and we hope to have applied each point in the best possible way. We are aware that this topic has not yet been widely explored through clinical studies and that therefore the limitations are the frequent use of paper on animal models but we hope to have aroused interest.

Reviewer 3 Report

Comments and Suggestions for Authors

The manuscript „Maternal Microbiome and Neurodevelopment: The Impact of Pregnancy Dysbiosis on Offspring Health” is well-structured, progressing logically from the introduction to the discussion and conclusions.

Need to revised:

Add numerical or statistical results where available to strengthen the findings.

The first paragraph introduces the concept of microbiota but should briefly explain why neurodevelopment is specifically affected by maternal dysbiosis.

The term “holobiont” is introduced without explanation—clarify its relevance to maternal-fetal health.

The statement “pregnancy dysbiosis is poorly investigated” should be supported with more references.

Adding a PRISMA diagram to the search strategy to illustrate the study selection criteria would improve transparency.

Specify whether the quality of the studies (e.g., assessment of risk of bias) was evaluated.

Avoid redundancy in listing databases; streamline phrases such as “databases such as PubMed, Scopus, and Google Scholar, including research published from January 2000 to January 2025.”

Some sections, such as “Gastrointestinal Microbiota”, contain extensive background information that would be better placed in the introduction or discussion. Consider relocating it.

Key findings from the reviewed studies should be highlighted with statistical data where possible.

The paragraph discussing microbiota changes in low- and middle-income countries (LMICs) should clarify how environmental enteric dysfunction (EED) is directly linked to neurodevelopmental impairment. Additional references are needed to support this claim.

Reducing redundancy in the definition of maternal immune activation (MIA) will improve readability.

The “two-hit theory” should be explained more clearly, as it is currently difficult to understand.

The section on dietary interventions is relevant, but references to specific dietary patterns should be supported by clinical trials where available. There are numerous studies on dietary patterns—please include additional references.

In the discussion on probiotic supplementation, specify which strains have been most effective in maternal populations.

The conclusion effectively highlights the importance of maternal microbiome research but should explicitly mention future directions, such as precision medicine approaches.

Avoid rhetorical questions in the conclusion section. They are more appropriate for the discussion or perspectives section.

This manuscript provides a compelling and insightful review of the impact of the maternal microbiome on neurodevelopment. By refining the structure, clarifying key findings, and ensuring concise scientific communication, it will be a valuable contribution to the field.

Author Response

Comments 1: Add numerical or statistical results where available to strengthen the findings.

Response 1: Thank for your comment. We added more study to strengthen the findings. 

Comment 2:The first paragraph introduces the concept of microbiota but should briefly explain why neurodevelopment is specifically affected by maternal dysbiosis.The term “holobiont” is introduced without explanation—clarify its relevance to maternal-fetal health.

Response 2: Thanks for your valuable point of view. We were proceeded to  going more into the depth of the findings.

Comment 3: The statement “pregnancy dysbiosis is poorly investigated” should be supported with more references.

Response 3: Thanks for your comment. We added more references to support the statement. 

Comment 4: Adding a PRISMA diagram to the search strategy to illustrate the study selection criteria would improve transparency.Specify whether the quality of the studies (e.g., assessment of risk of bias) was evaluated.Avoid redundancy in listing databases; streamline phrases such as “databases such as PubMed, Scopus, and Google Scholar, including research published from January 2000 to January 2025.”

Response 4: the type of the review is a Narrative review we were proceded to discuss more in details material and methods. 

Comments 5: Some sections, such as “Gastrointestinal Microbiota”, contain extensive background information that would be better placed in the introduction or discussion. Consider relocating it.

Response 5: Thanks for your suggestion, we have proceeded to relocate a part in the introduction.

Comments 6: The paragraph discussing microbiota changes in low- and middle-income countries (LMICs) should clarify how environmental enteric dysfunction (EED) is directly linked to neurodevelopmental impairment. Additional references are needed to support this claim. Reducing redundancy in the definition of maternal immune activation (MIA) will improve readability. The “two-hit theory” should be explained more clearly, as it is currently difficult to understand.The section on dietary interventions is relevant, but references to specific dietary patterns should be supported by clinical trials where available. There are numerous studies on dietary patterns—please include additional references. In the discussion on probiotic supplementation, specify which strains have been most effective in maternal populations.

Response 6: in this parts of the result and discussion we ware proceded to correct the parts requested. 

Comments 7: the conclusion effectively highlights the importance of maternal microbiome research but should explicitly mention future directions, such as precision medicine approaches. Avoid rhetorical questions in the conclusion section. They are more appropriate for the discussion or perspectives section.

Response 7: thanks for your comments. We were rewrote the conclusion. 

Round 2

Reviewer 2 Report

Comments and Suggestions for Authors

The rebuttal letter appears to lack specific details. It would benefit the Author to provide a more comprehensive response, addressing each point individually. This method enhances clarity and ensures that reviewers or readers can fully grasp how each concern or comment has been considered. A systematic approach can effectively showcase the Author's commitment to refining their work and openness to constructive feedback. I can see many items have been addressed but not properly mentioned in the Authors’ response.
1. Maternal Microbiome and Neurodevelopment: The Impact of Pregnancy Dysbiosis on Offspring Health
The title should identify if the study reports (human or animal) trial data or is a systematic review, meta-analysis, or replication study. https://www.mdpi.com/journal/nutrients/instructions#preparation
I am unsure whether this is a systematic or narrative review, although the Abstract's Methods suggest that it is a narrative review. This has not been addressed.
2. The Introduction addresses multiple aspects but may improve in terms of structural coherence. The shifts between concepts occasionally appear sudden, hindering readers' grasp of the overarching narrative.
This has not been addressed.
I suggest the Authors start with a paragraph on the gut microbiota, then the impact of pregnancy on gut microbiota, then the importance of investigating dysbiosis during pregnancy, then impact of urbanization and modern diet on microbial diversity and health, and finally, the interplay of microbiota in maternal and fetal health with the aims.
3. Introduction line 60. Please add the year of publication after “Contreras A. and Herrera JA, et al.” I said previously in round 1:I think that after “et al.” there should be a year of publication in parentheses. Please revise the entire manuscript.
4. Introduction. Line 79- the citation format is odd “[13-14-15-16]. “ Why not [13-16]?
5. Methods-how was bias going to be minimized?
6. Results—As I mentioned before, please do not use phrases such as “we know” (line 126).
7. Discussion. The references format is still wrong- e.g., “[33, 34, 35, 36]”, “[38-39-40]”. I said previously, “In the text, reference numbers should be placed in square brackets [ ], and placed before the punctuation; for example [1], [1–3] “. Please revise the entire manuscript.
8. Discussion- Please do not use phrases such as “we know” (line 321).
9. Discussion- The spacing in lines 236-246 is incorrect.
10. Discussion-In lines 315-319 “Moreover, Carole Brosseau et al., demonstrated how inulin and galactooligosaccharides (GOS) supplementation during gestation increased the Bacteroidetes:Firmicutes ratio in favor of Bacteroidetes, resulting in increased production of acetate detectable in amniotic fluid and following prebiotic supplementation, B and T cells increased in the fetus and remaining also in later stages of life [92].” What data is this: murine/human? Also, as I asked previously, this section does not clarify which data are human data. But I can see in Conclusions “Many studies are only on animal models.“ and I accept this.
11. Discussion- “figure 1.”should be capitalized, line 248.

Author Response

Comments 1: Maternal Microbiome and Neurodevelopment: The Impact of Pregnancy Dysbiosis on Offspring Health
The title should identify if the study reports (human or animal) trial data or is a systematic review, meta-analysis, or replication study. https://www.mdpi.com/journal/nutrients/instructions#preparation
I am unsure whether this is a systematic or narrative review, although the Abstract's Methods suggest that it is a narrative review. This has not been addressed.

Response: Dear reviewer, thank you for the comment. We proceeded to add the review type to the title.

Comments 2: The Introduction addresses multiple aspects but may improve in terms of structural coherence. The shifts between concepts occasionally appear sudden, hindering readers' grasp of the overarching narrative.This has not been addressed. I suggest the Authors start with a paragraph on the gut microbiota, then the impact of pregnancy on gut microbiota, then the importance of investigating dysbiosis during pregnancy, then impact of urbanization and modern diet on microbial diversity and health, and finally, the interplay of microbiota in maternal and fetal health with the aims.

Response: Thanks for your suggestion. we proceeded to improve the introductory part.

Comments 3:  Introduction line 60. Please add the year of publication after “Contreras A. and Herrera JA, et al.” I said previously in round 1:I think that after “et al.” there should be a year of publication in parentheses. Please revise the entire manuscript.

Response: thanks for the comment, the authors did not include the year of the work in the text as they have reviewed the Nutrient standards in which it is not required to do what is said above.

Comments 4:  Introduction. Line 79- the citation format is odd “[13-14-15-16]. “ Why not [13-16]?

Response: We have reported all the references taken into consideration in full.

Comments 5: Methods-how was bias going to be minimized?

Response: We proceeded to point out how biases were reduced during analysis and writing.

Comments 6:  Results—As I mentioned before, please do not use phrases such as “we know” (line 126).

Response: We proceeded with the correction.

Comments 7: Discussion. The references format is still wrong- e.g., “[33, 34, 35, 36]”, “[38-39-40]”. I said previously, “In the text, reference numbers should be placed in square brackets [ ], and placed before the punctuation; for example [1], [1–3] “. Please revise the entire manuscript.

Response: We proceeded with the correction.

Comments 8:  Discussion- Please do not use phrases such as “we know” (line 321).

Response: We proceeded with the correction.

Comments 9: Discussion- The spacing in lines 236-246 is incorrect.

Response: We proceeded with the correction.

Comments 10: Discussion-In lines 315-319 “Moreover, Carole Brosseau et al., demonstrated how inulin and galactooligosaccharides (GOS) supplementation during gestation increased the Bacteroidetes:Firmicutes ratio in favor of Bacteroidetes, resulting in increased production of acetate detectable in amniotic fluid and following prebiotic supplementation, B and T cells increased in the fetus and remaining also in later stages of life [92].” What data is this: murine/human? Also, as I asked previously, this section does not clarify which data are human data. But I can see in Conclusions “Many studies are only on animal models.“ and I accept this.

Response: We proceeded to specify that the study was carried out on mouse models. thanks for the correction.

Comments 11: Discussion- “figure 1.”should be capitalized, line 248.

Response: Thank you. We proceeded with the correction.